# The incubation history of soil samples strongly affects the occlusion of particulate organic matter

Frederick Büks[1], Sabine Dumke[1], Julia König[1]

[1]Chair of Soil Science, Dept. of Ecology, Technische Universität Berlin, 10587 Berlin, Germany

*Correspondence to:* Frederick Büks (frederick.bueks@tu-berlin.de)

**Abstract.** Soil structure is a key proxy for carbon and nutrient storage, stable pore space and rootability. It is often quantified based on the degree of aggregation or the mechanical stability of soil aggregates. This work compares two methods representing basic principles of aggregate measurement. Undisturbed soil samples of loamy sand, clayey silt and silty loam were analyzed by ultrasonication/density fractionation (USD) to quantify different soil organic
carbon (SOC) pools and by wet-sieving to measure the amount of water stable aggregates (%WSA). The measurements were carried out on field-fresh soils at field capacity (pF 1.8) as well as samples that were air-dried, reset to pF 1.8 by capillary action and incubated for 0, 1 and 4 weeks. Our results show, that the strength of POM occlusion sharply decreases after rewetting, indicated by the reduction of the more strongly bound occluded carbon fraction.
The respective amounts decreased by -4.5 wt% for loamy sand, -6.8 wt% for clayey silt as well as -16.3 wt% for silty loam, and the field fresh values are not fully recovered within the following four weeks. In contrast, the amount of water stable aggregates (%WSA) remains largely stable except in clayey silt, that shows an increase by +5.9 wt% directly after rewetting. In consequence, field-fresh measurements are highly recommended to avoid
overestimation of free and weakly bound SOM fractions or the degree of aggregation.

## 1 Introduction

It has been over 90 years since soil structure began to be in the focus of agricultural research (Russell, 1928; Christensen, 1930). First seen under the aspect of soil plowability, the 1960s brought attention on well aggregated soils as a support for root growth and against soil compaction by heavy machinery (Rosenberg, 1964). Today, good soil structure is seen as an eminent proxy of soil quality, as it does not only provide rootability and stable pore space, but is also related to water holding capacity, drainage of excess rain or flood water, enhanced aeration as well as carbon and nutrient storage within aggregates (Bronick and Lal, 2005).

The quantification of soil aggregates was early performed by use of dry and wet sieving (Yoder, 1936; Chepil and Bisal, 1943). From here, the methods branched out into approaches still used to describe the amount and size distribution of soil aggregation. One branch comprised the weight fraction of water-stable aggregates (%WSA) and, based on the same wet-sieving principle, the mean weight diameter (Bryant et al., 1948; Bavel, 1950; Angulo et al., 2024; Meidl et al., 2024). The other branch was aimed to measuring soil structure for its mechanical stability. Wet sieving methods were early applied to quantify the mechanical integrity of soils (aggregate stability) after a certain amount of stress provided by sieve movements (Russell and Feng, 1947). In the second half of the 20th century, ultrasonic dispersion introduced by Edwards and Bremner (1967a) increasingly replaced sieving methods for quantification of aggregate stability, since now the energy input to the soil could be estimated from the power output of the sonotrode (North, 1976). Nowadays, ultrasonication is a common tool to achieve a semi-quantitative view on soil structural stability by comparing the mass of aggregate size fractions or mean weight diameter of water stable aggregates (WSA) after the application of defined quantities of ultrasonic power ($J\ ml^{-1}\ s^{-1}$) (e.g. Lehtinen et al., 2014; Jouquet et al., 2016; Cavael et al., 2020). Beyond that, ultrasonication is combined with density fractionation (USD) of particulate organic matter (POM), which is successively released with increasing energy input and used for the quantification of soil carbon pools (Edwards and Bremner, 1967b; Golchin et al., 1994; Kaiser and Berhe, 2014; Graf-Rosenfellner et al., 2016).

However, carbon pool measurements that apply USD extraction are prone to a number of artifacts. A density cut-off at 1.6 $g\ cm^{-3}$ is mandatory, because lower concentrated solutions might not completely separate the respective POM fraction, while higher densities cause co-extraction of the mineral matrix (Cerli et al., 2012). Ultrasound treatments with energy levels >50 $J\ ml^{-1}$ can cause comminution of POM, sorption to mineral surfaces and lead to a reduced recovery rate, as demonstrated by Büks et al. (2021). This causes a carry-over of POM from fractions with lower to those with higher binding strength and a false estimation of both. Furthermore, adding the dense solution to the soil sample results in a low recovery rate of the free POM fraction (fPOM) due to burying within the matrix (Büks, 2023). This problem is addressed by rinsing the sample into the solution or gentle rotation, which significantly increases the recovery of fPOM and reduces overestimation of occluded POM (oPOM) mass. If the samples are air-dried, the abrupt addition of water or dense solutions to the soil causes rupture of WSA and release of oPOM by a process called slaking (Emerson, 1967; Bossuyt et al., 2001), which could in turn result in an overestimation of the fPOM fraction. In consequence, comparability of fractionation results is only given under standardized test conditions, that represent natural conditions of the soil sample as good as possible.

The present work focuses on another parameter, that can potentially cause carbon pool artifacts: The wetting history of the sample. Normally measuring under field-fresh conditions, the extent of a sampling campaign or samples from soil archives can make it necessary to

quantify structural characteristics of already air-dried soil. This may have influence on the measured soil structural parameters. Aggregate stability is increasing from low to high soil moisture as e.g. shown with sieving experiments (Liu et al., 2025) and rainfall simulators (Martínez-Mena et al., 1998). Air-drying, however, can increase the mechanical stability of soil aggregates by precipitation of various inorganic (and organic) cementation agents (Amézketa, 1999) and, potentially, the transfer of dissolved organic matter (DOM) from outer to inner spherical binding patterns (Kaiser et al., 2015).

Another factor of aggregate stability, that is not only influenced by the actual water content, but also its near history, is the soil microbiome and its biofilm matrix. The composition and activity of soil bacterial and fungal communities adapt to the soil water content, and bacterial rather than fungal abundance is significantly reduced under dry conditions (Drenovsky et al., 2004; Chowdhury et al., 2011). Since biofilms consist of 90-99 wt% water, they reshape during the drying process suggesting altered mechanical strength. The air-drying and long-term storage of soil samples therefore may alter the biofilm matrix and fungal hyphae networks, that have been shown to raise the quantity and stability of soil aggregates (e.g. Büks and Kaupenjohann, 2016; Bossuyt et al., 2001; Tang et al., 2011).

This, however, implies a decline of soil structural stability due to air-drying and long-term storage, when samples are rewet, compared to field-fresh measurements. On results of both, USD and WSA measurements, the destabilization of soil aggregates should have a similar effect, since acting as nucleus in aggregate formation POM is widely bound to the mineral matrix (e.g. Witzgall et al., 2021) and POM and MAOM carbon fractions are correlated with WSA (Bouajila and Gallali, 2010, Veum et al., 2012). Past studies on soil structural stability, POM occlusion and the amount of WSA have been conducted with either field-fresh, air-dried and re-moistened soil samples (e.g. Oztas and Fayetorbay, 2003; Annabi et al., 2007; Büks and Kaupenjohann, 2016). The aim of this work is to elucidate effects of this different practices on the comparability of the resulting data. We hypothesize that air-drying of soil samples causes the weakening of soil structure expressed by the decline of parameters such as percentage by weight of WSA and the POM occlusive strength, and that this effect can be attenuated by long enough re-incubation under conditions of field capacity.

## 2 Material and methods

### 2.1 Sites and sample preparation

Undisturbed moist soil samples were taken in late March 2024 at three different organic farming sites in Eastern Germany within a homogeneous area of each 1 m² (Table 1). The organic litter and top 10 cm were removed, and sampling was carried out in 10-20 cm depth of the mineral topsoil by use of metal rings ($\varnothing_i$=5.6 cm, h=4.0 cm, V=100 ml, n=25 per site). The rings containing the soil core were capped and tightly packed in closed plastic bags to preserve soil humidity for transpiration. Additionally, SOC concentrations of the sites were determined by each five mixed samples (each by three drillings) of the upper 30 cm topsoil along the bed.

**Tab. 1:** Field characteristics of the three sampling sites. For additional information, see Supplements.

| Texture | SOC (g kg$^{-1}$ dry mass) | pH | Region | Coordinates |
|---|---|---|---|---|
| Loamy sand (Sl2) | 32.2 ± 1.3 | 5.9 ± 0.1 | Brandenburg/Germany | 52°46'59.7"N 13°11'55.7"E |
| Clayey silt (Ut3) | 19.7 ± 2.1 | 7.1 ± 0.1 | Saxony-Anhalt/Germany | 51°10'38.9"N 11°57'27.6"E |
| Silty loam (Lu) | 18.4 ± 2.2 | 6.9 ± 0.1 | Thuringia/Germany | 50°53'14.2"N 11°12'42.7"E |

### 2.2 Setting to field capacity

Back in the laboratory, ten of the cores of each soil were set on a porous plate, saturated via capillary action for 24 hours and then drained by a hydrostatic head of pF 1.8 until constant weight was achieved. Half of these cores were weighed, dried for 24 hours at 105°C, weighed again, and the field capacity was determined from the difference minus the ring weight. The other five were directly measured (called "field-fresh" treatment) for soil organic carbon (SOC) fractions and percentage by weight of water stable aggregates (%WSA). The remaining 3x5 cores were air-dried at 35°C with strong air circulation until constant weight was reached and then set to pF 1.8 as described above. Five of these were directly measured for occlusion of POM carbon and %WSA (W0), the others were incubated at constant water content and 20°C in the dark for 1 and 4 weeks and then treated similarly.

### 2.3 fPOM, oPOM$_{50}$ and residual SOC

For analysis of SOC fractions (Fig. 1), small metal rings ($\varnothing$=2.4 cm, h=4.0 cm, V=18.1 ml) were used to subsample approx. 20 g dry soil equivalent from each of the field-fresh and re-incubated soil cores (n=5). The samples were gently removed from the ring by use of a spatula, weighted and rinsed into 200 ml Polyethylene (PE) flasks with 100 ml of sodium polytungstate solution (SPT) following Büks (2023). As the soils differ in their field capacity, the density of the SPT solution was adapted to match 1.6 g cm$^{-3}$ after addition to the sample by following Eq. 1

$$\frac{1.6\,g\,cm^{-3}}{100\,ml} = \frac{\rho_{SPT}}{100\,ml + m \cdot V} \quad \text{(Eq. 1)}$$

with m the dry mass of the soil sample (g), V the volume of soil solution per g soil (ml g$^{-1}$) and ρ$_{SPT}$ the required density of added SPT solution. Subsequently, the sample was stored at room temperature for 30 min to allow infiltration of SPT solution into the pore space.

The samples were then centrifuged for 26 min at 3,500 g. The floating fPOM was separated by use of a water-jet pump, filtered through a 0.45 µm cellulose acetate filter and cleaned with deionized water until the electrical conductivity of the filtrate dropped below 50 µS cm$^{-1}$. The samples were flushed with deionized water though a 2 mm sieve to remove coarse material and into aluminum bottles, freezed at -20 °C, lyophilized, dried for 24 h at 105 °C and stored in an exsiccator with a desiccant battery.

After separation of the fPOM, the PE flasks were refilled to their initial weight with 1.6 g cm$^{-1}$ dense SPT solution. Cavitational stress was applied to the samples by use of a sonotrode (Branson© Sonifier 250, sonotrode diameter of 13 mm, frequency of 40 kHz, immersion depth of 15 mm, power output of 50.02 ± 1.29 J s$^{-1}$) as described by Büks and Kaupenjohann (2016). Sonication time corresponding to the applied energy density of q=50 J ml$^{-1}$ was determined based on the energy output of the sonotrode calculated following North (1976). The subsequent extraction and preparation of the respective fraction (oPOM$_{50}$) was conducted similar to the fPOM.

The residuum (R), containing the oPOM releasable with q>50 J ml$^{-1}$ and the mineral-associated organic matter (MAOM), was washed approx. 6 times with 100 ml deionized water until the solution dropped below 50 µS cm$^{-1}$ or showed no further decrease of conductivity by washing. Then, these samples were treated similar to the fPOM and oPOM$_{50}$ fractions.

All fPOM, oPOM$_{50}$ and R samples were weighted, ground and the SOC concentrations were determined by use of an Elementar UNICUBE® CNS analyzer.

## 2.4 Water stable aggregates

The %WSA were quantified in each soil and treatment by use of a wet sieving apparatus (Eijkelkamp) following DIN 19683-16 and the manual instructions (Fig. 1, Eq. 2) (Kemper and Rosenau, 1986; Vrána et al., 2024). Similar to the above preparation, undisturbed subsamples of about 4 g dry weight equivalent were taken from the five cores of each treatment by use of a smaller metal ring ($\varnothing$=1.6 cm, h=1.7 cm, V=3.4 ml). The soil was carefully transferred from the ring to a 250 µm as above and the sieve was placed into a beaker with 90 ml of deionized water.

The samples were sieved for 3 min with a frequency of 34 min$^{-1}$. The sievings remained within the first beaker, while the sieve was placed into a new one filled with 70 ml of deionized water. The submerged sample was treated with ultrasound by use of a sonotrode (50.02 ± 1.29 J s$^{-1}$) for 2 min (~60 J ml$^{-1}$) to fully disaggregate all macroaggregates. Then, 20 ml of deionized water were added and the samples were sieved again as described above. The matter remaining within the sieve (R) was wet-sieved by use of a 2 mm mesh to remove coarse material. The sievings of the first (m$_1$) and second step (m$_2$) as well as R were dried for 24 h at 105 °C and set to room temperature in an exsiccator with a desiccant battery.

The mass fraction m$_1$ represents primary particles and microaggregates with diameters <250 µm, that were not part of water-stable macroaggregates, whereas m$_2$ represents all objects <250 µm that were released through the ultrasonication of water-stable macroaggregates. The fraction R is the sum of objects >250 µm, that did not pass the mesh during both steps of separation. The fraction

$$x_{WSA} = 100\% \cdot \frac{m_2}{m_1 + m_2 + R} \quad \text{(Eq. 2)}$$

with $m_2$ the matter <250 µm of water stable aggregates, m1 the matter <250 µm of water labile aggregates and free primary particels and R the residuum of matter between 250 and 2000 µm allows for semi-quantitative measurement of water stable macroaggregates. Particles >2 mm are excluded from the assessment, as samples can differ greatly in their coarse fraction due to small stones and litter.

**2.5 Statistics**

A pre-search with 1.5 interquartile range method (1.5xIQR) was applied to identify potential outliers within the SOC and %WSA data sets and the regular data of each treatment were tested for normal distribution by use of the Shapiro-Wilk test ($\alpha=0.05$). Since all WSA and SOC data were normally distributed, Grubbs' test for outliers was applied ($\alpha=0.05$) despite of the low number of replications (n=5). A few outliers were identified across the whole data, that, however, do not affect the significance of the data (see supplements). Data of %WSA and SOC fractions were also positively tested for homogenity of variance (Levene's test, p>0.05). For each soil separately, the data were compared by Student's t-test between the field-fresh and each of the incubation treatments. In addition, linear regression analyses of the %WSA with the SOC fraction data were carried out.

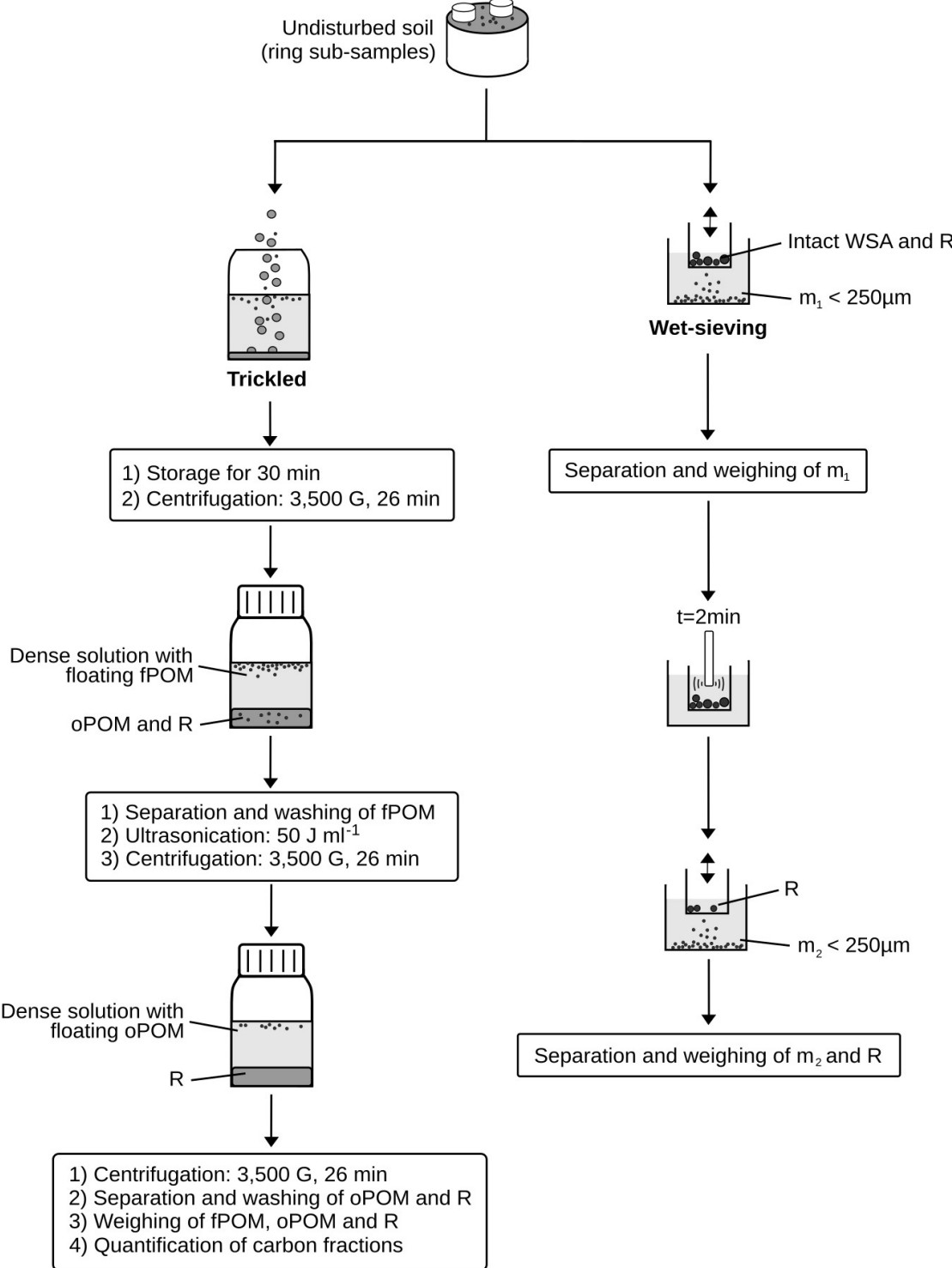

**Fig. 1:** Process chart of SOC pool (left) and water-stable macroaggregate measurement (right). Abbreviations refer to free (fPOM) and occluded particulate organic matter (oPOM), non-aggregated particles <250 μm ($m_1$), particles <250 μm of water-stable macroaggregates ($m_2$) and the residua of both methods (R).

# 3 Results

## 3.1 Shifts of SOC fractions

Directly after being rewet (0 weeks), the soils consistently show increased amounts of the fPOM-C and $oPOM_{50}$-C fractions as well as corresponding decreases of the residual fractions, compared with the field-fresh treatments (Fig. 2). This shift is significant ($p<0.05$) for all samples except the fPOM of clayey silk ($p=0.22$) and the $oPOM_{50}$ of loamy sand ($p=0.06$). The loamy sand thereby shows the largest increase of fPOM (+3.3 wt%) directly after being rewet, which mainly corresponds to the decrease of the residual fraction. The clayey silt and the silty loam, in contrast, have a rather small release of additional fPOM, and the decrease of the residual fraction corresponds with the $oPOM_{50}$ fraction (+5.8 wt% and +14.8 wt %,respectively).

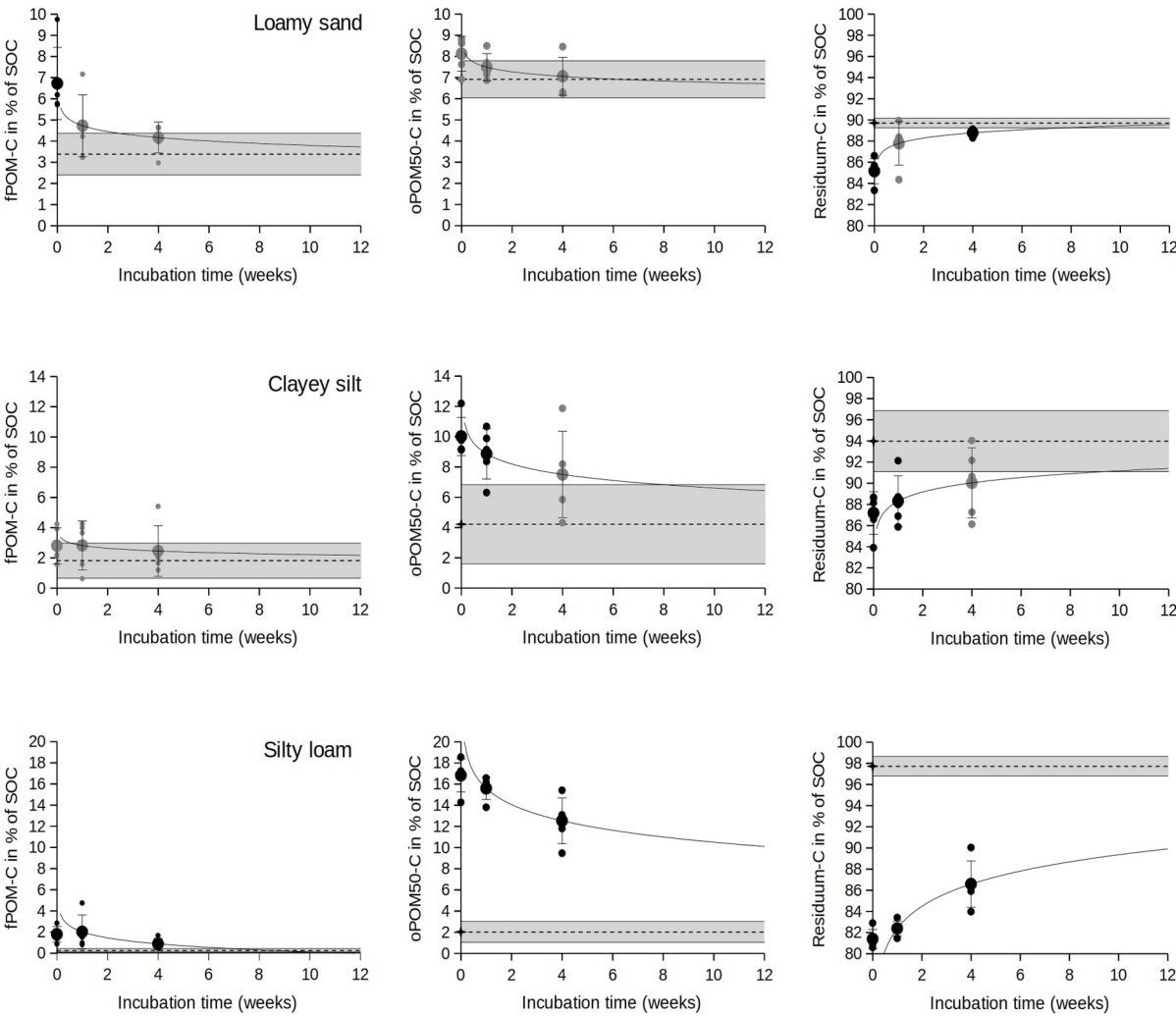

**Fig. 2**: Development of the fPOM-C, $oPOM_{50}$-C and residual C fraction of a loamy sand, a clayey silt and a silty loam after drying and re-incubation for 0, 1 and 4 weeks. The gray bar refers to the respective values of the field-fresh samples (dotted=mean value, outer lines=standard deviation). Large dots are mean values, small dots single measurements. Black dots refer to treatments with values significantly different from the field-fresh samples, gray marks similarity ($\alpha=0.05$). The logarithmic fits illustrate the trend towards the field-fresh values with full lines within and dotted lines beyond the measured span.

All samples, furthermore, show a consistent trend towards the initial values of the field-fresh
samples. The loamy sand reaches non-significant differences to field-fresh values in the
fPOM and $oPOM_{50}$ fraction within 1 week (t-test p≥0.05). The clayey silt takes 4 weeks in all
SOC fractions, while the silty loam does not restore in any fraction.

**3.2 Alteration of water stability and correlation with SOC fractions**
Of the three soil types, only the clayey silt shows significant (p<0.05) increases of %WSA
directly after re-incubation and returns to non-significant differences compared to the field-
fresh values within 1 week (Table 1). The loamy sand have a tendency for increased %WSA
directly after being rewet. All soils thereby show higher variation compared to the above SOC
measurements (1.4 wt%), with mean standard deviations of 4.2 wt% for the heavy soils and
9.1 wt% for the sand. The residual fractions of the respective soils have nearly constant mean
values across all treatments differing by <1.6 wt% (see supplements).

**Tab. 2:** Development of the amount of water stable macroaggregates ($m_2$, <250 µm), unbound particles ($m_1$, <250 µm) and residual particles (R, 250–2000 µm) of loamy sand, clayey silt and silty loam in field-fresh state as well as after drying and re-incubation for 0, 1 and 4 weeks. The * refers to significant differences compared to the field-fresh samples (Student's t-test, p<0.05).

| | | field-fresh | 0 weeks | 1 week | 4 weeks |
|---|---|---|---|---|---|
| **Loamy sand** | $m_1$ | 34.0 ± 12.7 | 27.4 ± 5.6 | 38.4 ± 8.6 | 40.5 ± 10.4 |
| | $m_2$ | 51.8 ± 14.1 | 57.7 ± 4.8 | 48.7 ± 8.6 | 46.9 ± 11.6 |
| | R | 14.2 ± 1.6 | 15.0 ± 3.8 | 12.9 ± 1.7 | 12.6 ± 3.8 |
| **Clayey silt** | $m_1$ | 16.8 ± 2.5 | 11.7 ± 4.3 | 13.8 ± 4.7 | 14.3 ± 6.3 |
| | $m_2$ | 78.9 ± 2.8 | 84.8 ± 4.3* | 82.7 ± 4.7 | 81.7 ± 6.2 |
| | R | 4.3 ± 0.8 | 3.6 ± 0.2 | 3.5 ± 0.3* | 4.0 ± 1.0 |
| **Silty Loam** | $m_1$ | 6.6 ± 2.9 | 7.6 ± 4.0 | 7.8 ± 2.5 | 4.1 ± 1.1 |
| | $m_2$ | 86.3 ± 2.9 | 85.3 ± 4.0 | 85.6 ± 3.4 | 89.3 ± 1.3 |
| | R | 7.1 ± 1.2 | 7.1 ± 0.7 | 6.6 ± 1.0 | 6.6 ± 0.7 |

The regression of %WSA ($m_2$) with $oPOM_{50}$-C plus residual C (the amount of SOC, that
should be occluded within water stable aggregates) shows no correlation, neither for loamy
sand ($R^2$=0.16) nor for clayey silt ($R^2$=0.10) and silty loam ($R^2$=0.11). Regression of $m_1$ with
the fPOM-C and %WSA with only the $oPOM_{50}$-C or only the residual C lead to similar results
($R^2$<0.21, $R^2$<0.04 and $R^2$<0.16, respectively).

**4 Discussion**

Our results show, that POM occlusion in sandy, silty and loamy soils is substantially decreased by air-drying and subsequent rewetting. The effect diminishes over time of re-
incubation, but did not restore field-fresh values of all SOC fractions within a span of four weeks. Longer periods of incubation were avoided in this experiment, since they increase the risk, that the isolated soil ecosystem within the sampling ring is overgrown by fungal hyphae most probably affecting aggregate characteristics (Tisdall et al., 1997; Fan et al., 2022).

The decreased strength of POM occlusion is indicated by an increased mass of the fPOM-C
and weakly bound $oPOM_{50}$-C fraction as well as the corresponding decrease of the residual C fraction, that contains the more strongly bound oPOM and the MAOM. This implies an initial section by section shift from the residual C to the $oPOM_{50}$-C and further to the fPOM-C fraction as a result of air-drying and rewetting. The sandy loam, which has generally lower aggregate stability due to its low clay content, thereby shows a stronger increase of the
fPOM-C fraction, compared to the heavier soils. This can be interpreted as partial destruction of soil aggregates causing release of POM. In contrast, clayey silt and silty loam have strong shifts from the residual to the $oPOM_{50}$-C fraction, which accounts for the majority of destabilized POM and indicates a weakening of soil aggregate structure without release of POM. Having decreasing mechanical stability of soil structure in common, all three soils show
different pattern of destabilization with mainly destruction (loamy sand), mainly weakening (clayey silt) and both destruction and weakening. In field studies, the observed weakening of soil structure after reincubation will lead to an overestimation of fPOM and the loosely occluded POM fraction, while the respective residuum is underestimated. This causes overestimation of the labile C pool and underestimation of the binding forces within
aggregates. This underpins the need for USD measurements with field-fresh samples.

In contrast to POM occlusion, the percentage by weight of water stable aggregates is less affected by air-drying and gentle rewetting. Only the clayey silt (and in tendency the loamy sand) shows a significant increase of water stable aggregates directly after re-incubation with a subsequent restoration of field-fresh values, while the other treatments have largely
constant state. Deviations within the residuum and a significant increase of the clayey silt residual fraction after 1 week can be better explained by a different composition of the 250–2000 µm mineral and organic matter fraction than by incomplete dispersion of macroaggregates. The widely constant amount of water stable aggregates matches results of Mikha et al. (2005), who observed constant aggregate size classes after dry-wet cycling. This
is in contrast to different works that show a loss of macroaggregates after mainly fast rewetting events (e.g. Denef et al., 2001; Bossuyt et al., 2001). Haynes and Swift (1990) compared field-fresh and air-dried soil samples after wet-sieving and found the mean weight diameter of macroaggregates in grassland soils to be increased, while arable soils showed decreased values compared to field-fresh samples. Due to the rare number of studies on this
topic, there is only speculation, weather and to which extent these opposite observations are caused by soil texture, SOM quantity and quality as well as methodology.

The observed lack of correlation with the SOC measurements seems contradictory at first, as it could be assumed that an unchanged amount of WSA implies unchanged oPOM fractions and, thus, *decreasing* the amount of water-stable macroaggregates causes an increase of
fPOM-C at the expense of the oPOM fraction. However, both methods use incongruent parameters, as shown by Almajmaie et al. (2017), who stated poor correlation between different methods of soil structure measurement. The WSA approach represents the ratio of soil particles <250 µm of water-stable macroaggregates to the total sample mass. If

measurements are conducted with samples set to pF 1.8, shear forces of wet sieving are the
prominent agent of dispersion. In contrast, the USD measurement provides the amount of fPOM, that is fractionated by trickling through SPT solution containing approx. 1.2 M $Na^+$ as chemical dispersion agent, and oPOM fraction that is released from aggregates under certain levels of cavitational stress in the same $Na^+$ rich environment. With both methods addressing different binding mechanisms and using different forces of disaggregation, results may,
arguably, diverge. As an example, drying could weaken but not fully destroy the structural integrity of existing soil aggregates by dehydration of biofilms and fungal hyphae, that work as important aggregation agents in soils, and cause an increased release of weakly bound oPOM after the USD treatment, while the amount of non-aggregated material passing the mesh in the WSA measurement remains constant as shown for silty loam. Air-drying was also
shown to cause extensive death of the soil microbiome and rewetting can induce rapid mineralization of unprotected SOC such as biofilm components, known as the "Birch effect" (Birch, 1958, Kaiser et al., 2015, Schroeder et al., 2021). This may alter the binding pattern of POM within soil aggregates. As a result of a dehydration treatment, the increase of %WSA may occur through the formation of slightly soluble precipitates, that influence soil structural
stability directly after rewetting. This points out, that also WSA measurements should be carried out with field-fresh samples to avoid e.g. overestimation of the amount of soil aggregates of soils. If changes of structural integrity are associated with altering the size of water stable macroaggregates, they can be addressed by measuring the mean weight diameter (MWD) as extension of the %WSA method (Kemper and Rosenau, 1986).
From a mechanistic perspective, drying and rewetting are not clearly attributed to positive or negative influence on aggregate stability, since different effects such as the loss of OM as a binding agent, partial damage of biofilms, cementation and the evolution of SOM-mineral interaction from outer to inner-spheric binding pattern might work in opposite directions. Also magnitude and duration of the respective effects after rewetting are still not estimated and
require further studies, which should also include shrinkage and swelling of clay-rich soils or the span of storage under air-dried conditions.

In consequence, studies on soil structure in mostly humid regions should generally use field-fresh samples. If there is no option except storing the soil samples, defined re-incubation should be carefully applied for ≥1 week. Furthermore, Kühnel et al. (2019) showed that similar
masses of SOM fractions were measured in soil samples both air-dried and frozen for long-term storage, which makes freezing an additional option of storage, if also microbial analyses are on the schedule. In regions with dry seasons, however, severe droughts during the summer months and punctual raining events are regular phenomena. Most of the time, the topsoil is close to air-dry, and rainfall sharply increases the soil water content, likely leading to
aggregate breakdown. If analyzing WSA, POM occlusion or aggregate geometry in such cases, fast rewetting without subsequent incubation, even with the acceptance of slaking, can be suitable to simulate natural conditions properly. On the other hand, slow moisturing by capillary action is indicated, if soil structure needs to be preserved. From our point of view, rinsing dry soil aggregates into SPT solution should be avoided, to prevent enhanced slaking
due to high ionic strength of dissolved $Na^+$ similar to the solution of sodic soils (Rengasamy and Olsson, 1991; Liu et al., 2021). If comparing samples from different sites, that are taken in different seasons, with different cropping or weather history, the influence of seasonally changing aggregate stability (e.g. Tian et al., 2023) as well as underlying factors such as adapting microbiome (McDaniel and Grandy, 2016, Kim et al., 2020) should be taken into
account with regard to the respective research question.

**5 Conclusion**

The present study shows, that measuring structural characteristics of soils with very different soil textures is strongly affected by drying and re-incubation treatments. The strength of POM occlusion decreases after rewetting, even if slaking is avoided, and is not recovered within the following four weeks. In contrast, the amount of water-stable macroaggregates remains stable and only increased significantly in one soil directly after rewetting. This work shows the importance of soil structure measurements with field-fresh samples to avoid overestimation of free and weakly occluded POM fractions and water stable aggregates. The structure of measurement campaigns should be adapted to that issue. It further underpins that both investigated methods differ in their measured characteristics and should be used together and not substitute each other.

**Data availability**

All of the data are published within this paper and in the Supplement.

**Author Contributions**

FB developed the experimental concept, organized and conducted the soil sampling and the laboratory work, analyzed the data and prepared the manuscript. SD planned the extraction process, adapted the USD method, conducted and managed the experiment. JK conducted the laboratory work and data collection.

**Competing interests**

The authors declare that they have no conflict of interest.

**Acknowledgements**

Many thanks go to Sandra Reimann, Carlotta Kollmann and Christine Beusch for their laboratory support. We are also very grateful to the farmers for the opportunity to sample their soils and the German Federal Ministry of Education and Research for funding (BMBF, funding number 01UU2202).

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
