# Peer review of "The incubation history of soil samples strongly affects the occlusion of particulate organic matter"

_EGUsphere, 2025_

## Author Response (AR1)

Dear referee #1.

Thank you very much for your helpful comments, especially the recommendation to consider measurements in soils of semiarid and arid regions. In the following I want to answer your comments to the best of our knowledge. Within the revised manuscript, you will find the corrections based on your comments marked up with green, the comments of referee#2 marked up with purple and my own points with pink.

Best regards,

Frederick Büks

**Line 28.** *Suppress 'but' at the end of the line (repeated word).*
→ Thank you very much. Deleted.

**Line 45.** *'...application of defined quanta of ultrasonical stress'. What is a 'quanta' of ultrasonical stress? I suspect that you meant that a defined level or strength of energy (Watts) or physical work (Joules) is applied to the sample. If I am right, I suggest you put it in this way, easier to understand. If I am wrong, then try to be more clear about the concept ('quanta').*
→ Yes, very right. We will replace the quanta by "quantities of ultrasonic power" ($J\ ml^{-1}\ s^{-1}$).

**Line 62.** *'And be there an air-dried sample'. Is it a direct translation from German? Dou you mean, simply 'If the soil sample is air-dried, the abrupt addition...'.*
→ Changed to "If the samples are air-dried, …"

**Line 101.** *Below the term 'pH' there is a dot between parentheses '(·)'. What does it mean? If unnecessary, I suggest to remove it.*
→ It's actually a dash symbolizing "no SI unit here". I deleted it to avoid misinterpretations.

**Lines 301-304.** *This final comment is of interest. Authors are german, thus used to soils that rarely undergo a severe drought. But for a mediterranean like me, the extreme drought of summer months is a usual phenomenon, not an artifact of laboratory. To me the physical fractionation directly on dry samples may be an opportunity to study the reconfigurations happened in the soil (the precise architecture of particles, WSA, POM particles) during drought. May I suggest you mention this aspect of the problem.*
→ Thank you. We added "in humid regions" in Line 296 and expanded the paragraph: "In regions with dry seasons, however, severe droughts during the summer months and punctual raining events are regular phenomena. Most of the time, the topsoil is close to air-dry, and rainfall sharply increases the soil water content, likely leading to aggregate breakdown. If analyzing WSA, POM occlusion or aggregate geometry in such cases, fast rewetting without subsequent incubation, even with the acceptance of slaking, can be suitable to simulate natural conditions properly. On the other hand, slow moisturing by capillary action is indicated, if soil structure needs to be preserved. From our point of view, rinsing dry soil aggregates into SPT solution should be avoided, to prevent enhanced slaking due to high ionic strength of dissolved $Na^+$ similar to the solution of sodic soils (Rengasamy and Olsson, 1991; Liu et al., 2021)."

**FIGURES.** Be careful with Fig. 2. Letters are very small: it may become illegible in the printed version. Even in the screen version, I had to increase the view to 150 to read the precise meaning of the Y-axis.
→ We increased the font size.

Dear referee #2.

Thank you very much for your helpful comments. They focus on points that are very valuable for the content of the work. In the following I want to answer your comments to the best of my knowledge. All corrections based on your recommendations are marked within the revised manuscript with purple, those of referee#1 with green and my own points with pink.

Best regards,

Frederick Büks

In the beginning, I want to correct a misconception: The present work do not aim to assess the influence of different water contents on the occlusion of SOM, but the respective effect of different times of re-incubation at a fixed pF value (see section 2.2). The "field fresh" control, in contrast to the other treatments, thereby refers to "not air-dried after sampling", and these samples were set to pF 1.8 similar to the other treatments. We are sorry for the misunderstanding and will try to clarify this within the abstract and the methodological description e.g. by adding "field capacity (pF 1.8)" in L16 and the corrections below. Due to the fixed pF value, your comment on seasonal effects of different water contents is omitted, but aspects of it are addressed by our reply to referee#1.

*I was surprised that the moisture content of fresh aggregates is not mentioned (did I miss something?), and that the authors chose to rewet the aggregates to pF = 1.8 rather than to the moisture content observed in the field. The rationale behind this decision should be clarified.*

→ Thank you for that point. We add the moist state and sampling month to L94. Although the exact soil water content is not known for all samples, this underlines that the samples have been already near field capacity and setting field fresh samples to field capacity directly after sampling will not have led to slaking or similar artifacts.

*Moreover, the fact that soil aggregates are remoistened after drying is not discussed in relation to potential microbial responses. One could reasonably expect a rapid mineralization of unprotected carbon following rewetting, which may alter the distribution of POM within aggregates by the end of the incubation period.*

→ This is indeed one of the prominent effects assumed to make re-incubation practice difficult. We added the following to L283: "Air-drying was also shown to cause extensive death of the soil microbiome and rewetting can induce rapid mineralization of unprotected SOC such as biofilm components, known as the "Birch effect" (Birch, 1958, Kaiser et al., 2015, Schroeder et al., 2021). This may alter the binding pattern of POM within soil aggregates." and "From a mechanistic perspective, drying and rewetting not clearly attributed to positive or negative effect on aggregate stability, since different effects such as the loss of OM as a binding agent and the evolution of SOM-mineral interaction from outer to inner-spheric binding pattern might work in opposite directions (Kaiser et al., 2015)." to L290.

**L10**: *The abstract would benefit from the inclusion of key quantitative findings or indicative figures. Presenting numerical evidence is essential.*

→ That's right. We added "The respective amounts decreased by -4.5 wt% for loamy sand, -6.8 wt% for clayey silt as well as -16.3 wt% for silty loam and the field fresh values are …" (L19) and "…, that shows an increase by +5.9 wt% directly after rewetting." (L20) to the abstract.

**L22**: *The term "erosion stability" requires clarification. Aggregate stability is not synonymous with erosion resistance; other processes such as crust formation, detachment, and transport must also be considered when discussing erosion. Please revise accordingly.*
→ Thank you. We removed it from L11, replaced it in L22 by "degree of aggregation", in L287 by "the amount of soil aggregates" and in L312 by "water stable aggregates".

**L70**: *The statement on the relationship between soil moisture and aggregate stability is questionable. Generally, higher soil moisture can lead to greater aggregate stability due to reduced air-filled porosity and decreased slaking upon wetting. However, the interpretation provided in the cited article appears to relate to shear strength, which is conceptually and mechanistically distinct from water-stable aggregation. Please ensure that the citation is appropriate and revise the interpretation accordingly.*
→ We replaced the sentence in L70 by the following. "This may have influence on the measured soil structural parameters. Aggregate stability is increasing from low to high soil moisture as e.g. shown with sieving experiments (Liu et al., 2025) and rainfall simulators (Martínez-Mena et al., 1998). Air-drying, however, can increase the mechanical stability of soil aggregates by precipitation of various inorganic (and organic) cementation agents (Amézketa, 1999) and, potentially, the transfer of dissolved organic matter (DOM) from outer to inner spherical binding patterns (Kaiser et al., 2015)."

**L73:** *Consider adding a statement acknowledging that aggregate stability varies seasonally, often independently of current soil moisture content. There is substantial evidence that dry aggregates sampled at different times of the year can exhibit markedly different stability due to biological and physico-chemical changes. This important temporal dynamic is currently overlooked and should be addressed with appropriate references.*
→ Yes. We added "If comparing samples from different sites, that are taken in different seasons, with different cropping or weather history, the influence of seasonally changing aggregate stability (e.g. Tian et al., 2023) as well as underlying factors such as adapting microbiome (McDaniel and Grandy, 2016, Kim et al., 2020) should be taken into account with regard to the respective research question." to L304.

**L105:** *There is a lack of coherence between the content of this paragraph and the information presented in Figure 1. The text mainly discusses how moisture values for pF 1.8 were obtained and the number of samples per treatment, whereas Figure 1 appears to focus on the methodology for extracting OM and WSA. Please revise for consistency and clarity.*
→ Yes, the reference to Fig. 1 does not belong there. Removed.

**L152:** Please add a reference that employs this methodology.
→ The apparatus was used very early by Kemper and Rosenau (1986) (Aggregate stability and size distribution. *Methods of soil analysis: Part 1 Physical and mineralogical methods*, 5, 425-442.) and recently applied e.g. by Vrána et al. (2024) (A laser diffractometry technique for determining the soil water stable aggregates index. Geoderma, 441, 116756). Added to L153.

**L174:** *Equation 2 is not explained. Additionally, in the results section, values derived from this equation appear to have been multiplied by 100.*
→ Thank you. We added "with $m_2$ the matter <250 µm of water stable aggregates, m1 the matter <250 µm of water labile aggregates and free primary particels and R the residuum of matter between 250 and 2000 µm" to L171 and corrected the equation.

***L180:*** *Please specify the software used for statistical analyses, the generation of plots.*
→ Statistics and Plots are generated by LibreOffice Calc. The gray background layer in Fig. 2 was added manually.

***L212:*** *The phrasing is unclear and should be reworded for better comprehension.*
→ Sorry, this is really a messy sentence. Removed, because the fact that the gain of aggregated material is almost equal to the loss of non-aggregated material has no informational value in this case.

***L218–221:*** *The content reads more like a discussion than a results section.*
→ Moved to L260.

***L262:*** *How do the authors explain this apparent contradiction? It would also be important to state whether it is common to find a relationship between POM and WSA.*
→ To address your first question, we add "Due to the rare number of studies on this topic, there is only speculation, weather and to which extent these opposite observations are caused by soil texture, SOM quantity and quality as well as methodology." to L264.
→ The second point is addressed in L83 by "On results of both, USD and WSA measurements, the destabilization of soil aggregates should have a similar effect, since acting as nucleus in aggregate formation POM is widely bound to the mineral matrix (e.g. Witzgall et al., 2021) and POM and MAOM carbon fractions are correlated with WSA (Bouajila and Gallali, 2010, Veum et al., 2012)."